# RANKL Inhibition Reduces Cardiac Hypertrophy in *mdx* Mice and Possibly in Children with Duchenne Muscular Dystrophy

**DOI:** 10.3390/cells12111538

**Published:** 2023-06-03

**Authors:** Laetitia Marcadet, Emma Sara Juracic, Nasrin Khan, Zineb Bouredji, Hideo Yagita, Leanne M. Ward, A. Russell Tupling, Anteneh Argaw, Jérôme Frenette

**Affiliations:** 1Centre Hospitalier Universitaire de Québec, Centre de Recherche du Centre Hospitalier de l’Université Laval (CHUQ-CHUL), Axe Neurosciences, Université Laval, Quebec City, QC G1V 4G2, Canada; laetitia.marcadet@gmail.com (L.M.); zineb.bouredji@crchudequebec.ulaval.ca (Z.B.); anteneh.argaw@crchudequebec.ulaval.ca (A.A.); 2Department of Kinesiology and Health Sciences, University of Waterloo, Waterloo, ON N2L 3G1, Canada; e.s.juracic@gmail.com (E.S.J.); rtupling@uwaterloo.ca (A.R.T.); 3The Ottawa Pediatric Bone Health Research Group, Children’s Hospital of Eastern Ontario Research Institute, Ottawa, ON K1H 8L1, Canada; nakhan@cheo.on.ca (N.K.); lward@cheo.on.ca (L.M.W.); 4Department of Immunology, School of Medicine, Juntendo University, Tokyo 113-8421, Japan; hyagita@juntendo.ac.jp; 5The Department of Pediatrics, University of Ottawa, Ottawa, ON K1H 8L1, Canada; 6Department of Rehabilitation, Université Laval, Quebec City, QC G1V 0A6, Canada

**Keywords:** RANK, RANKL, heart, Duchenne muscular dystrophy, cardiac hypertrophy

## Abstract

Cardiomyopathy has become one of the leading causes of death in patients with Duchenne muscular dystrophy (DMD). We recently reported that the inhibition of the interaction between the receptor activator of nuclear factor κB ligand (RANKL) and receptor activator of nuclear factor κB (RANK) significantly improves muscle and bone functions in dystrophin-deficient *mdx* mice. RANKL and RANK are also expressed in cardiac muscle. Here, we investigate whether anti-RANKL treatment prevents cardiac hypertrophy and dysfunction in dystrophic *mdx* mice. Anti-RANKL treatment significantly reduced LV hypertrophy and heart mass, and maintained cardiac function in *mdx* mice. Anti-RANKL treatment also inhibited NFκB and PI3K, two mediators implicated in cardiac hypertrophy. Furthermore, anti-RANKL treatment increased SERCA activity and the expression of RyR, FKBP12, and SERCA2a, leading possibly to an improved Ca^2+^ homeostasis in dystrophic hearts. Interestingly, preliminary post hoc analyses suggest that denosumab, a human anti-RANKL, reduced left ventricular hypertrophy in two patients with DMD. Taken together, our results indicate that anti-RANKL treatment prevents the worsening of cardiac hypertrophy in *mdx* mice and could potentially maintain cardiac function in teenage or adult patients with DMD.

## 1. Introduction

Duchenne muscular dystrophy (DMD), which mostly affects males, is an X chromosome-linked recessive genetic disorder that prevents the synthesis of a functional dystrophin protein. Dystrophin is mainly responsible for anchoring the actin cytoskeleton in muscle fibers to the extracellular matrix. The absence of dystrophin weakens the sarcolemma, making it more susceptible to damage and resulting in progressive skeletal muscle degeneration/regeneration cycles [1] and cardiac dysfunction later in life [2]. Intracellular calcium (Ca^2+^) plays a central role in numerous cellular processes. A deficiency in dystrophin leads to a loss of Ca^2+^ homeostasis and to skeletal and cardiac muscle dysfunctions [3]. In the early stages, an elevated intracellular Ca^2+^ concentration in the hearts of *mdx* mice increases the mitochondria Ca^2+^ uptake capacity as a compensatory phenomenon, in response to sarcoplasmic reticulum (SR) dysfunction [4]. Ultimately, the consistent increase in Ca^2+^ disrupts the polarization of the mitochondrial membrane, which is necessary for the production of ATP, and leads to their death [5]. Of note, cardiac ion channels regulating potassium and sodium are also affected by the absence of dystrophin [6]. The dysregulation of cardiac ion channels and the resulting mitochondrial dysfunction occur prior to the development of cardiomyopathy [6].

Dystrophin-deficient cardiomyocytes are more susceptible to contraction-induced injury [7] and, once damaged, they cannot regenerate, limiting the potential for cardiac muscle repair. Cardiomyocyte dysfunction in DMD causes fibrosis and the loss of contractile components, leading to heart failure and the premature death of patients in their third decade of life [8,9].

The receptor activator of nuclear factor κB (RANK), the receptor activator of nuclear factor κB ligand (RANKL), and the osteoprotegerin (OPG) signaling pathway play a central role in bone homeostasis. RANKL, which is part of the tumor necrosis factor (TNF) superfamily, is secreted by osteoblasts while RANK, its receptor, is located on pre-osteoclastic cells. The RANKL/RANK interaction induces the formation of multinucleated mature osteoclasts, ultimately leading to bone resorption and remodeling [10]. OPG, the third component, is produced by osteoblasts and binds to RANKL, inhibiting RANKL/RANK interactions [10]. Although it was suggested by Julius Wolff almost two centuries ago that skeletal muscles and long bones form a functional unit [11], the signaling pathway that synchronously governs the two tissues during development, aging, and disease is still not well understood. Recent work from our laboratory and others have shown that the RANK/RANKL/OPG triad is more than just a bone regulator, and also plays a role in muscle diseases [12,13,14]. The inhibition of the RANKL/RANK interaction with denosumab, a human monoclonal antibody targeting RANKL, increases muscle strength in a mouse model of sarcopenia and in post-menopausal women with osteoporosis [15]. It has recently been reported that treatment with anti-RANKL also reduces cachexia in mice with non-metastatic ovarian cancer [16]. We have shown that full-length OPG fused with Fc (FL-OPG-Fc), anti-RANKL treatment, and muscle RANK deletion can each improve, to different degrees, skeletal muscle function in dystrophic mice [17]. Several studies have reported the expression of RANKL and RANK in cardiac tissue [13,18,19]. Interestingly, the structural remodeling of the myocardium in pathological hearts involves the RANK/RANKL/OPG triad [13]. Anti-RANKL therapy has been shown to reduce infarct size and cardiac neutrophil infiltration in C57BL/6 mice [20]. Furthermore, we have also uncovered that inhibition of the RANKL/RANK interaction by the muscle-specific deletion of RANK enhances force production and modulates intracellular Ca^2+^ dynamics, sarco(endo)plasmic reticulum Ca^2+^-ATPase (SERCA2a) expression, and SERCA activity in denervated fast-twitch muscles [21]. SERCA2a is the primary isoform expressed in cardiac muscle. It is a key regulator of muscle performance and Ca^2+^ homeostasis [22]. A dysregulation in Ca^2+^ handling leads to the activation of Ca^2+^-dependent proteases, such as muscle-specific calpains, and gives rise to a pathological cycle of degeneration and chronic inflammation [23]. This evidence clearly demonstrates that the RANK/RANKL/OPG pathway plays an important role in skeletal and cardiac muscle diseases [14].

Male *mdx* mice, the animal model most commonly used for studying DMD, share several features with the DMD-associated cardiomyopathy observed in dystrophic boys [24]. For example, abnormal electrocardiograms (ECG) and echocardiograms have been reported in *mdx* mice [25,26]. Furthermore, the changes in cardiac function are associated with an increase in myocardial fibrosis, the appearance of necrotic foci, and inflammation of the myocardium [24,27]. Nevertheless, cardiac dysfunctions remain poorly understood in dystrophic *mdx* mice, and several studies using mice of different ages have reported conflicting results. We thus characterized cardiac function in aging male *mdx* mice and investigated the pathophysiological roles played by the RANKL/RANK interaction with respect to dystrophic heart morphology and function. Here we report that dystrophic *mdx* mice developed the first signs of pathological cardiac hypertrophy, detectable by echocardiography, at 11 months of age. Our results also show that the anti-RANKL treatment markedly reduced myocardium hypertrophy and increased SERCA activity in the hearts of dystrophic mice. Interestingly, an exploratory post hoc analysis we performed reveals that treatment with denosumab, a human monoclonal antibody targeting RANKL, appears to have reduced cardiac hypertrophy in two children with DMD.

## 2. Materials and Methods

All analyses and experiments were performed by a blind experimenter. All the data generated and analyzed during the present study are included in this article.

### 2.1. Animals

Male wild-type C57BL/10J (WT) and dystrophic C57BL/10ScSn-*Dmd^mdx^*^/*J*^ (*mdx*) mice were purchased from the Jackson Laboratory (Bar Harbor, ME, USA) and were bred at our animal facility. Dystrophic *mdx* mice were injected intraperitoneally every 3 days for 2 months starting at 11 months of age, with either a vehicle (PBS: phosphate-buffered saline) or anti-RANKL (4 mg/kg/3d) [28]. We have previously shown that an anti-RANKL treatment of 4 mg/kg/3d is more effective than a dose of 1 mg/kg/3d [28]. Furthermore, we did not see any additional beneficial effects with an anti-RANKL treatment dose of 8 mg/kg/3d [28]. WT mice were used as the control. At the end of the experimental procedures, the mice were euthanized by cervical dislocation under anesthesia with an intraperitoneal injection of sodium pentobarbital (Euthanyl^®^ 120 mg/kg). The hearts were removed and stored at −80 °C for future analysis.

### 2.2. Echocardiography

It is challenging to obtain non-invasive images of the hearts of animals, such as mice, due to their small size and very fast heart rate. Therefore, echocardiography was performed once the mice were anesthetized with 2.5% isoflurane inhalation to reduce stress and to maintain a heart rate ranging from 475 to 525 bpm [29], which minimizes variations in measurements. We used a 21 MHz ultrasound probe and a Vevo^®^ 2100 System (Visual Sonics, Toronto, ON, Canada). All the ultrasound measurements were recorded in our small animal imaging facility at the CRCHU. To limit analysis bias, all the images were acquired and analyzed by a “blind” experimenter using the papillary muscles as a reference point [30]. High-quality images with an adequate delineation of the endocardial borders were used for hemodynamic analyses. The small-axis parasternal view using M-mode imaging was recorded to measure the anterior and the posterior left ventricular (LV) wall thickness and the LV internal diameter (Appendix A). The mass of the LV, the ejection fraction (EF), and the fractional shortening (FS) were calculated using the built-in Vevo LAB analysis software (Visual Sonics, software version 5.6, Toronto, ON, Canada).

### 2.3. Heart Histology

Transverse sections (10 µm) were cut from the isolated hearts using a Leica cryostat (Leica Microsystems CM1850, Concord, ON, Canada) and fixed in 4% paraformaldehyde solution. Tissue section autofluorescence was quenched with a solution composed of 155 mM NaCl, 10 mM Tris-HCl (pH 7.5), and 100 mM NaBH_4_. The tissue sections were washed three times with PBS, and then were blocked with 2% BSA and 1% horse serum in PBS. The tissue sections were incubated overnight at 4 °C with the following antibodies: anti-RANK (ab13918, Abcam, Toronto, ON, Canada), anti-RANKL (NBP2-61813, Novus Biological, Centennial, CO, USA), anti-F4/80 (MCA479, Serotec, Neuried, Germany), or anti-CD206 (ab64693, Abcam, Toronto, ON, Canada), and anti-laminin (NB300144, Novus Biological, Centennial, CO, USA or L9393, Sigma-Aldrich, Oakville, ON, Canada) diluted in blocking buffer. The sections were washed three times with PBS and were then incubated with Alexa Fluor 488- or Alexa Fluor 594-conjugated secondary antibodies (Invitrogen), or rhodamine-phalloidin (R415, Thermo Fisher Scientific, Saint-Laurent, QC, Canada), for 2 h at room temperature. The sections were washed three times with PBS and were mounted on slides with Fluoromount-G^TM^ containing 4′, 6-diamidino-2-phenylindole (DAPI). Images of the immunofluorescence staining were acquired using a ZEN2 AxioCam camera (Zeiss, North York, ON, Canada). Other sections were stained with Masson’s trichrome to detect collagen. An image of the entire section was acquired using a ZEN2 AxioCam camera. All the images were analyzed using ImageJ (software version 1.53). The cross-sectional areas of 500 cardiomyocytes located in separate regions from five non-consecutive sections along the apex and the papillary muscle axis of each heart were measured. Macrophages were counted from nine non-overlapping fields along the apex and the papillary muscle axis of each heart.

### 2.4. Western Blotting

The hearts were homogenized in a lysis buffer containing 1 μg/mL of protease inhibitor cocktails (P8340, Sigma-Aldrich, Oakville, ON, Canada), 20 mM Tris-base (pH 7.5), 140 mM NaCl, 1 mM MgCl_2_, 1 mM CaCl_2_, 10% glycerol, 1% Igepal (I3021, Sigma-Aldrich, Oakville, ON, Canada), 2 mM Na_3_VO_4_, 8.3 mM NaF, and 0.2 mM PMSF. The protein content of the supernatant was measured using a BCA protein assay kit (EMD Chemical, Oakville, ON, Canada). The protein homogenates were separated on SDS polyacrylamide gels. The protein bands were transferred to nitrocellulose membranes (Nitro, Amersham, VWR, Mississauga, ON, Canada). The total transferred proteins were stained with Ponceau S. The membranes were then washed in PBS-T and blocked in 5% BSA. The membranes were cut to probe the different sized proteins and were incubated overnight at 4 °C with the following primary antibodies: anti-RANK (BAF692, R&D Systems, Toronto, ON, Canada), anti-RANKL (NBP2-61813, Novus Biological, Centennial, CO, USA), anti-pNFκB (3033, Cell Signaling, Whitby, ON, Canada), anti-NFκB (4764, Cell Signaling, Whitby, ON, Canada), anti-pPI3K (sc-293115, Santa Cruz Biotechnology, Dallas, TX, USA), anti-PI3K (sc-423, Santa Cruz Biotechnology, Dallas, TX, USA), anti-RyR (sc-13942, Santa Cruz Biotechnology, Dallas, TX, USA), and anti-FKBP12 (ab108420, Abcam, Toronto, ON, Canada). The membranes were washed in PBS-T and were incubated with the appropriate horseradish peroxidase (HRP)-conjugated secondary antibodies (Santa Cruz Biotechnology, Dallas, TX, USA or Abcam, Toronto, ON, Canada). Bands were revealed using Immobilon Western Chemiluminescent HRP Substrate (Millipore Sigma, Oakville, ON, Canada). Images of the membranes were acquired with a LAS-1000plus Luminescent Image Analyzer (Fujifilm, Mississauga, ON, Canada) and were analyzed using Quantity One software (v4.6.6, Bio-Rad, Mississauga, ON, Canada). The optical density of each target protein was normalized to the optical density of the total proteins on a Ponceau S-stained membrane. The results were expressed as a fold change relative to WT.

### 2.5. SERCA Activity

The frozen heart samples were homogenized 1:10 (*w*/*v*) in an ice-cold homogenizing buffer (250 mM sucrose, 5 mM HEPES, 0.2 mM PMSF, and 0.2% [*w*/*v*] NaN_3_) using a glass homogenizer. SERCA activity was measured in A. R. Tupling’s laboratory using an assay adapted for a spectrophotometer plate reader, and based on the oxidation of NADH at 340 nm in an assay buffer (pH 7.0) containing 1 mM EGTA, 10 mM phosphoenolpyruvate, 18 U/mL of pyruvate kinase and lactate dehydrogenase, 0.2 mM NADH, 20 mM HEPES, 200 mM KCl, 15 mM MgCl_2_, 10 mM NaN_3_, and 5 mM ATP. The homogenized muscles and assay buffer were added to tubes containing 16 different concentrations of Ca^2+^ (ranging from 7.6 to 4.7 pCa units) in the presence of 4.2 μM calcium ionophore A23187 (Millipore Sigma, Oakville, ON, Canada) to mitigate any back inhibition as a result of SR vesicle filling. NADH (0.3 mM) was then added to start the reaction, and the optical density was recorded at 340 nm for 30 min at 37 °C. The different concentrations of Ca^2+^ were used to determine the maximal enzyme activity (Vmax) and pCa50. Cyclopiazonic acid (CPA, 40 μM), a highly specific SERCA inhibitor, was used to determine background activity.

### 2.6. SERCA2a and PLN Western Blotting

The protein expression levels of SERCA2a and PLN were assessed using standard Western blotting techniques. The heart homogenates were solubilized in a 1× solubilizing buffer (0.1% 2-Mercaptoethanol, 0.0005% Bromophenol blue, 10% Glycerol, 2% SDS, and 63 mM Tris-HCl (pH 6.8)) and the proteins were separated using SDS-PAGE. Upon electrophoresis, the separated proteins were transferred onto a polyvinylidene difluoride (PVDF) membrane with a transfer buffer (25 mM glycine, 192 mM Tris base, 20% methanol, 0.1% [*w*/*v*] SDS). The membranes were then incubated in a blocking solution (TBST buffer: 20mM Tris base, 137 mM NaCl, and 0.1% (*v*/*v*) Tween 20, pH 7.5, with 5% [*w*/*v*] non-fat dry milk). The proteins were immunoprobed with the corresponding primary antibodies for SERCA2a ((2A7-A1, product no. MA3–919), monoclonal mouse antibody acquired from Thermo Fisher Scientific) and PLN ((2D12, product no. MA3-922) monoclonal mouse antibody acquired from Thermo Fisher Scientific, Saint-Laurent, QC, Canada). Subsequently, the proteins were immunoprobed with an HRP-conjugated secondary antibody: goat anti-mouse IgG, which was obtained from Santa Cruz Biotechnology, Dallas, TX, USA. Luminata Forte™ was used to detect SERCA2a and PLN. GeneTools (Software version 4.3, Syngene, Frederick, MD, USA) was utilized to quantify the resulting optical densities. All the values were normalized to protein content as determined from Ponceau staining.

### 2.7. Children with DMD

In collaboration with Dr. Leanne Ward (University of Ottawa, Ottawa, ON, Canada), four boys with DMD underwent echocardiography before, and two years after, the start of zoledronic acid (n = 2) or denosumab (n = 2) as part of a clinical trial for the treatment of bone fragility due to osteoporosis. The height and weight of the patients were measured before each echocardiography. LV hypertrophy is usually defined by its mass index (LVMI, g/m^2^). Since children are short in stature and the LVMI includes a normalization by height, the LVMI value tends to increase during childhood. Given this, LV mass-for-height centile curves are more accurate than LVMI as a method for normalizing LV mass to body size in children [31]. The four children affected were treated for bone fragility due to osteoporosis and received either a subcutaneous injection of the human anti-RANKL antibody denosumab (1 mg/kg every 6 months), or intravenous bisphosphonate zoledronic acid (Zol, 0.025 mg/kg every 6 months). 

### 2.8. Statistical Analyses

All the values are expressed as means ± SEM. The data were analyzed using Student’s t-test, a one-way ANOVA with Tukey’s post hoc test, or a two-way ANOVA with a Bonferroni correction (GraphPad Prism software version 9.5). The level of significance was set at * *p* < 0.05, ** *p* < 0.01, *** *p* < 0.001, or **** *p* < 0.0001.

## 3. Results

This section will be divided by subheadings. It will provide a concise and precise description of the experimental results and their interpretation, as well as the experimental conclusions that can be drawn.

### 3.1. Dystrophic Mice Develop Left Ventricular Hypertrophy and Exhibit a Decrease in Cardiac Function

Fractional shortening (FS), corresponding to the variation in the wall thickness between the diastole (relaxation) and systole (contraction) of dystrophic and healthy hearts, varies considerably between studies. For example, Quinlan et al. in 2004 [24] and Spurney et al. in 2008 [32] reported a FS of 45% [24] and 23.5% [32], respectively, in 10-month-old *mdx* mice and a FS of 58% and 33.6%, respectively, in their age-matched controls. The internal diameter and thickness of the myocardium, two important indicators of cardiac function, were also very different in these two studies [24,32]. Van Putten et al. (2014) [33] and Fayssoil et al. (2013) [26] also reported conflicting results for the ejection fraction of 10-month-old *mdx* mice. Given the discordant results reported in the literature on the appearance of cardiac dysfunction in dystrophic *mdx* mice [24,26,32,33], we evaluated myocardium morphology and cardiac function at three different ages. Echocardiography was performed on 10-, 11-, and 13-month-old male mice (Figure 1A and Appendix A). *Mdx* and age-matched wild-type (WT) mice have a similar body mass at 10, 11, and 13 months of age (Figure 1B). The LV wall thickness in diastole, measured from the echocardiography, was significantly higher in the 10-month-old *mdx* mice compared to the age-matched WT mice (Figure 1C). A thicker ventricular wall is considered one of the first signs of heart enlargement [34]. Accordingly, we observed an increase in LV wall thickness between 10 and 13 months in the WT and *mdx* mice (Figure 1C). The LV mass estimated by echocardiography was similar in the *mdx* and age-matched WT mice at 10 months of age, but was significantly higher in the 11- and 13-month-old *mdx* mice (Figure 1D). When normalized to body mass, the LV mass was significantly higher in the 11- and 13-month-old *mdx* mice compared to the age-matched WT mice (Figure 1E). We also observed an increase in LV mass and LV mass normalized to body mass with an age between 10 and 13 months in the WT and *mdx* mice (Figure 1D,E, respectively). The *mdx* mice exhibited a 46% increase in the LV mass to body mass ratio, while the gain was limited to 30% in the WT mice (Figure 1E).

The myocardial thickening observed at 10 months of age did not appear to have a deleterious effect on cardiac function in the *mdx* mice, as indicated by a LV ejection fraction (LVEF, Figure 1F) and LV fractional shortening (LVFS, Figure 1G) similar to those of the age-matched WT mice. However, cardiac function was significantly lower in the 13-month-old *mdx* mice, as shown by the decline in LVEF and LVFS compared to the age-matched WT mice (Figure 1F,G, respectively). A thickened left ventricle with an unchanged or increased LVEF are signs of adaptive compensatory hypertrophy [34]. Conversely, pathological cardiac hypertrophy is associated with cardiac dysfunction and is marked by a decrease in LVEF [34]. Overall, heart morphology evolves with age, as shown by the increase in the LV wall thickness (Figure 1C), the LV mass (Figure 1D), and the LV mass/body mass ratio (Figure 1E), between 10 and 13 months of age in both the WT and *mdx* mice. These modest morphological changes had no effect on the cardiac function of the WT mice (Figure 1F,G), while cardiac morphology and function worsened in the *mdx* mice, as demonstrated by the greater differences in LV mass/body mass ratio, LVEF, and LVFS between 10 and 13 months of age (Figure 1E–G, respectively).

### 3.2. An Anti-RANKL Treatment Improves Left Ventricular Morphology and Maintains Cardiac Function in Dystrophic Mice

We have previously shown that the RANK/RANKL/OPG pathway is involved in dystrophic skeletal muscle function [17,28] and is expressed in cardiac tissue [13,18,19]. To determine the effect of RANKL/RANK signaling on cardiac dysfunction in dystrophic mice, we first determined whether RANKL and RANK are expressed in dystrophic mouse hearts. The immunohistochemistry results show that RANKL and RANK are expressed in the microenvironment and on the cell membranes of the WT and dystrophic cardiomyocytes (Figure 2A,B, respectively). RANKL and RANK protein levels are significantly higher in the *mdx* hearts compared to the WT hearts (Figure 2C,D, respectively). 

To determine whether the inhibition of RANKL/RANK signaling affects dystrophic cardiac muscle function, the *mdx* mice were treated with an anti-RANKL for 2 months starting at 11 months of age, at the onset of cardiac dysfunction. The *mdx* mice were randomized into two experimental groups: PBS or anti-RANKL (4 mg/kg/3d) (Figure 3A and Appendix A). Although the body mass decreased between 11 and 13 months of age in the two experimental groups, it was similar in both groups at 11 (pre-treatment, Pre) and 13 months (post treatment, Post) of age (Figure 3B). The small loss in weight observed in the control and treatment groups was most likely due to the repeated injections (1 injection/3d, 18 injections in total), which may have been stressful for the mice. 

The LV wall thickness in diastole markedly increased between 11 and 13 months of age in the PBS-injected *mdx* mice (*mdx*-PBS, Figure 3C). At 13 months of age, the LV wall thickness in diastole of the anti-RANKL-treated *mdx* mice was significantly lower than that of the PBS-injected *mdx* mice (Figure 3C). A two-month anti-RANKL treatment appears to have reduced the LV wall thickness in the *mdx* mice (*mdx*-anti-RANKL Pre vs. Post, Figure 3C). The LV mass to body mass ratio increased significantly in the PBS-injected *mdx* mice from 11 to 13 months of age (Figure 3D), while a two-month anti-RANKL treatment markedly reduced this parameter (Pre vs. Post, Figure 3D). Cardiac contractility deteriorated in the PBS-injected 13-month-old *mdx* mice, as indicated by the significant decrease in LVEF and LVFS (Figure 3E,F, respectively). The anti-RANKL treatment prevented a decline in the LVEF and LVFS (Pre vs. Post, Figure 3E,F, respectively). In addition, at 13 months of age, the post-mortem heart mass expressed per body mass was significantly lower in the anti-RANKL-treated *mdx* mice than in the PBS-injected *mdx* mice (Figure 3G). These results demonstrate that the hearts of the anti-RANKL-treated *mdx* mice exhibited significantly improved LV morphology and better cardiac contractility.

### 3.3. An Anti-RANKL Treatment Reduces Cardiomyocyte Surface and Inhibits Cardiac Hypertrophy Mediators in Dystrophic Mice

Given the significant decrease in LV mass, we investigated whether the inhibition of the RANKL/RANK interaction affects cardiomyocyte morphology. First, we determined the effects of anti-RANKL on the cardiomyocyte cross-sectional area (CSA) and size distribution. Accordingly, the cardiomyocyte CSA was higher in the *mdx* mice (all treatment groups) when compared to their age-matched WT mice (Figure 4A).

### 3.4. An Anti-RANKL Treatment Increases SERCA Activity and Modulates Intracellular Calcium Homeostasis Regulators in Dystrophic Hearts 

A deficiency in dystrophin and the resulting instability of the sarcolemma have been associated with an increase in intracellular Ca^2+^ levels and cardiac dysfunction [3]. We previously reported that SERCA activity and expression levels are significantly reduced in *mdx* fast-twitch muscles compared to age-matched WT mice, suggesting that Ca^2+^ recapture in dystrophic skeletal muscles is dysregulated [17]. In the present study, we investigated whether SERCA activity is affected in dystrophic hearts and whether anti-RANKL modulates SERCA expression and function. Our results show that SERCA activity is significantly depressed in dystrophic hearts from a comparison of 13-month-old *mdx* mice to their age-matched WT (Figure 5A). 

Anti-RANKL administration completely rescues SERCA activity in dystrophic hearts (Figure 5A). SERCA2a protein levels were similar between the hearts from the *mdx*-PBS and their age-matched WT mice, while an anti-RANKL treatment markedly increased SERCA2a protein levels in the dystrophic hearts (Figure 5B). Phospholamban (PLN), an adaptor protein that inhibits SERCA activity, was increased in all the *mdx* groups of mice compared to the WT (PLN, Figure 5C). Anti-RANKL did not affect the expression level of PLN (Figure 5C). The loss of calcium homeostasis in DMD is also associated with the leakage of RyR and the depletion of its stabilizer FKBP12 [35]. The anti-RANKL-treated *mdx* mice exhibited increased RyR protein levels (Figure 5D), while no differences were observed in the WT and PBS-injected *mdx* mice. We thus investigated whether an anti-RANKL treatment would influence FKBP12 levels. Dystrophic hearts from the *mdx* mice displayed a lower expression of FKBP12 when compared to hearts from the WT mice (Figure 5E). Interestingly, FKBP12 protein levels increased significantly in the anti-RANKL-treated *mdx* mouse hearts compared to those from the PBS-injected dystrophic mice (Figure 5E). These results show, for the first time, that the inhibition of the Interaction between RANKL and RANK markedly rescues SERCA activity and the key regulators of Ca^2+^ homeostasis in dystrophic mouse hearts.

### 3.5. Denosumab Treatment Reduces Left Ventricular Hypertrophy in Patients with DMD

Four boys with DMD, three on deflazacort (Def; black, red, and yellow dots) and one on prednisolone (Pred, blue dots), were treated for approximately two years with either denosumab, a human monoclonal anti-RANKL (1 mg/kg/6 months, n = 2) or zoledronic acid (Zol), a bisphosphonate (0.025 mg/kg/6 months, n = 2) (Figure 6A) to minimize glucocorticoid (GC)-related bone loss. Denosumab and Zol are two FDA-approved antiresorptive treatments. Denosumab acts directly on the RANKL/RANK interaction while Zol blocks farnesyl pyrophosphate synthase, an enzyme important for osteoclast activity [36]. The age at the initiation of treatment varied from one patient to another, as did the dosage, which is based on body weight. Nevertheless, at the time of the second echocardiography, after treatment with denosumab or Zol, the GC to body mass ratios were similar among three of the patients (0.43 mg/kg for the black, red, and blue dots). One patient (yellow dots) had a higher GC to body mass ratio, 0.58 mg/kg (Figure 6A). At the beginning of the study, the denosumab-treated patients were 8 and 12 years old, while the Zol-treated children were 7 and 8 years old (Figure 6B). The children in the two groups were similar in terms of height, while their weight varied prior to and across the treatment period (Figure 6C,D, respectively). One Zol-treated child had a significant gain in weight during the two-year treatment period (Figure 6D, red dots). One denosumab-treated child presented with a high LV mass at the beginning of treatment (Figure 6E, yellow dots). However, once on denosumab, the LV mass declined by 36% (Figure 6E, yellow dots). The LV mass was maintained in the second child with the denosumab treatment (Figure 6E, blue dots). Interestingly, the LV masses of the Zol-treated children were slightly higher at the end of the treatment period (Figure 6E, black and red dots). The LV mass-for-height z scores, which express LV mass relative to the mean of healthy children of the same height, were above 0 for all four children before the treatments, suggesting that they showed signs of LV hypertrophy. Notably, the z scores for the LV mass of the denosumab-treated children diminished markedly below 0 and then remained constant, while they were above 0 in the Zol-treated children (Figure 6F). The ejection fractions ranged from 56 to 78% for all the children, which were normal [37], and remained constant during the treatments (Figure 6G), suggesting that the children did not appear to exhibit symptoms of systolic dysfunction during the observation period.

## 4. Discussion

The life expectancy of patients with DMD has significantly increased due to better management of the disease. Consequently, cardiomyopathy has become the leading cause of death in DMD patients [9]. Cumulative evidence supports the role of the RANK/RANKL/OPG triad in heart failure, infarcts, and cardiac remodeling [13,18,20]. Hypertrophied myocardium express and secrete RANKL, which induces pro-inflammatory cytokine production in a mouse model of pressure overload [19], while the expression of RANK, RANKL, and OPG increases following myocardial infarction [18]. Interestingly, the selective inhibition of RANKL in hematopoietic cells reduces the pro-inflammatory cytokine production and improves cardiac function after myocardial infarction [38]. In the present study, we report for the first time that RANKL/RANK signaling, a key bone regulator, is involved in dystrophic cardiac function. We show that dystrophic *mdx* mice exhibit cardiac hypertrophy and that the pharmacological inhibition of the RANKL/RANK interaction prevented the worsening of cardiac hypertrophy and upregulated SERCA activity and expression in dystrophic *mdx* mice. Intriguingly, post hoc analyses show that cardiac hypertrophy seemed to improve in the two patients treated with denosumab. While these are very preliminary observations, and one cannot state that the improvements are due to denosumab, the findings remain nevertheless provocative, given the favorable effects of anti-RANKL in the highly controlled murine *mdx* model.

Similar to patients with DMD, *mdx* mice develop cardiac dysfunction detectable by echocardiography at 10–13 months of age. Quinlan et al. (2004) [24], Zhang et al. (2008) [39], and Fayssoil et al. (2013) [26] showed that the hearts of *mdx* mice are hypertrophied, dilated, and fibrotic, and that they contract poorly [24,26,39]. In line with the aforementioned studies, our results indicate that the pathology progresses and cardiac function deteriorates gradually in aging *mdx* mice. In agreement with the results reported by Fayssoil et al. (2013) [26], our results show that 10-month-old *mdx* mice displayed early signs of cardiac hypertrophy which did not seem to affect systolic function. However, at 12 months of age, the *mdx* mice had a higher LV mass than the age-matched WT mice, and a reduced LV systolic function [26]. Our results indicate that the 13-month-old dystrophic mice exhibited a significant reduction in cardiac function, a sign of the transition from physiological cardiac hypertrophy to pathological cardiac hypertrophy [26,34]. Given that RANKL and RANK are expressed in dystrophic hearts and that *mdx* mice show signs of cardiac hypertrophy at 10 months of age, we investigated whether inhibiting RANKL would modulate cardiac hypertrophy and contractility in dystrophic mice. Interestingly, mouse antibody-mediated RANKL inhibition significantly slowed the progression of LV hypertrophy in the *mdx* mice. The LV mass to body mass ratio increased by 21% in the PBS-injected *mdx* mice, but decreased by 18% in the anti-RANKL-treated *mdx* mice. Consistent with these observations, the heart mass post mortem and the mean cardiomyocyte surface were significantly lower in the anti-RANKL-treated mice than in the PBS-injected *mdx* mice. The RANKL/RANK interaction activates downstream effector proteins such as NFκB and PI3K [40], which are involved in multiple signaling pathways. In the present study, the anti-RANKL treatment decreased pNFκB protein levels by 53% in dystrophic hearts, but very limited effects were seen with respect to cardiac fibrosis and inflammation (Appendix A). Although fibrosis and inflammation were very moderate in the hearts from 13-month-old *mdx* mice, our results are consistent with those of Slavic et al. (2018) [38], who showed that a 4 week anti-RANKL treatment following the permanent occlusion of coronary artery did not reduce the inflammation from myocardial infarction. Most importantly, the inhibition of the NFκB pathway by the conditional deletion of IKK in the cardiomyocytes rescued cardiac function in dystrophic mice without modifying fibrosis and inflammation [41]. The improvement in cardiac function was associated with an upregulation of calcium genes in the heart [41]. Furthermore, RANKL is a member of the TNF superfamily, and the TNF and RANKL signaling pathways considerably overlap. Previous studies have shown that TNF contributes to adverse LV remodeling and dysfunction during cardiac pressure overload [42], and that transgenic mice overexpressing TNF specifically in the heart develop cardiac hypertrophy [43]. In addition to NFκB, PI3K plays a major role in the pathogenesis of cardiac hypertrophy [44], and PI3K signaling is also an important pathway for RANKL-induced osteoclastogenesis [45,46]. Our findings show that pPI3K protein levels decreased by 26% in dystrophic hearts from the anti-RANKL-treated mice compared to those from the PBS-injected *mdx* mice. Thus, our results indicate that anti-RANKL injections suppress hypertrophic signaling, alleviating cardiac hypertrophy in the context of muscular dystrophy.

Normal cardiomyocyte function requires that the intracellular Ca^2+^ be sufficiently high in systole and low in diastole. Much of the Ca^2+^ needed for contraction comes from the SR membrane, while SERCA pumps cytosolic Ca^2+^ back into the SR during diastole. A persistent rise in the intracellular Ca^2+^ concentration is associated with poor cardiomyocyte contractility and compensatory cardiac hypertrophy [47]. SERCA2a expression and activity are reduced in many cardiac pathologies [48], while its activity is either normal or increases during physiological cardiac hypertrophy and decreases during the transition to pathological cardiac hypertrophy [34]. In the context of muscular dystrophy, SERCA activity is lower in 6-month-old *mdx* mice compared to age-matched WT mice [49], and this reduction is maintained in older *mdx* mice [50]. In agreement with these observations, our results indicate that there was a significant decrease in SERCA activity in the hearts of the 13-month-old dystrophic mice. Previous work from our laboratory has shown that the RANKL/RANK interaction modulates SERCA expression and activity in dystrophic skeletal muscles [17,21]. In line with our previous observations, the anti-RANKL treatment markedly enhanced SERCA activity and SERCA2a protein content, which may potentially improve Ca^2+^ homeostasis, impacting the progression of cardiac hypertrophy by enhancing cardiomyocyte contractility. Furthermore, we show that the pharmacological inhibition of RANKL maintained dystrophic cardiac function while it prevented a decline in the ejection fraction and the fractional shortening associated with pathological LV hypertrophy. Consistent with these findings, a single systemic delivery of human SERCA2a with an adeno-associated virus to 3-month-old *mdx* mice sustainably improved the ejection fraction when evaluated 18 months later [50]. It is also relevant to point out that dystrophic cardiomyocytes present with a Ca^2+^ leak through RyR2 channels due to the depletion of FKBP12 from the complex [35]. FKBP12 stabilizes the closed state of RyR [51] and reduces Ca^2+^ leakage during the diastolic phase [52]. Early treatment with a RyR2 stabilizer prevents an excessive SR Ca^2+^ leak and DMD-related cardiomyopathy in mice [53]. Here, we report that two months of anti-RANKL treatment also enhanced RyR and FKBP12 protein expression in the 13-month-old *mdx* mice. Collectively, our findings show that a 2-month anti-RANKL treatment beginning at 11 months of age was sufficient to increase RyR, FKBP12, and SERCA2a expression, and enhance SERCA activity, leading potentially to an improved Ca^2+^ homeostasis, a reduction in LV hypertrophy, and the maintenance of cardiac function in dystrophic hearts.

Based on these interesting results implicating the RANKL/RANK interaction in dystrophic myocardium integrity, and given that denosumab has been approved by the FDA for the treatment of osteoporosis, we investigated the possibility that our observations could be translated to patients affected by DMD. The post hoc echocardiography analyses we performed show that boys with DMD treated with denosumab experience decreases in LV hypertrophy. As with the dystrophic mice used in the present study, all four children showed early signs of LV hypertrophy at the beginning of the study, with an LV mass-for-height z score greater than 0. However, based on the clinical guidelines, which list a z score greater than 1.64, only one child had confirmed LV hypertrophy at this early age [31]. Most importantly, the z score for heart size dropped below zero in the denosumab-treated dystrophic children. Nevertheless, these results do not allow us to infer the direct effect of the denosumab, given the differences in the age at which the treatments were started, differences in the results at the baseline, differences in the dosage and the type of GCs used and, importantly, because of the limited number of patients available for these analyses. It is noteworthy to mention that a meta-analysis has indicated that denosumab therapy is not associated with any adverse cardiovascular effects in patients with primary osteoporosis, while romosozumab, a humanized sclerostin inhibitor that increases bone formation and decreases bone resorption, may increase the risk of major adverse cardiovascular events in osteoporotic patients, particularly in elderly men and postmenopausal women [54]. Our clinical observations thus suggest that denosumab could potentially reduce cardiac hypertrophy in patients with DMD, an observation that requires further study on a larger number of patients in a randomized, controlled study design. We speculate, based on our findings with *mdx* mice, that denosumab improves intracellular Ca^2+^ homeostasis in cardiomyocytes, possibly by increasing SERCA activity and/or by minimizing Ca^2+^ leakage from RyR.

## 5. Conclusions

In conclusion, the pharmacological inhibition of the RANKL/RANK interaction with anti-RANKL in a mouse model of DMD reduces the cardiac hypertrophy inherent to the disease and maintains systolic function. Post hoc, exploratory cardiac function analyses of patients treated with denosumab also align with the beneficial effects observed in dystrophic mice. We have previously shown that anti-RANKL has a beneficial effect on bone and skeletal muscle [17,28], and this study confirms its ability to limit heart hypertrophy in dystrophic mice. The effect of anti-RANKL on dystrophic heart hypertrophy most likely occurs through the upregulation and rescue of SERCA activity, which represents a promising strategy for treating DMD and DMD-induced cardiomyopathy [50]. The present study opens an interesting perspective from which a single drug may preserve the integrity of the three most affected tissues, the skeletal muscles, bones, and the heart, in DMD. The extent to which denosumab may prevent both LV hypertrophy and osteoporosis requires further study in the setting of a larger trial that is sufficiently powered to assess these outcomes more definitively.

## Figures and Tables

**Figure 1 cells-12-01538-f001:**
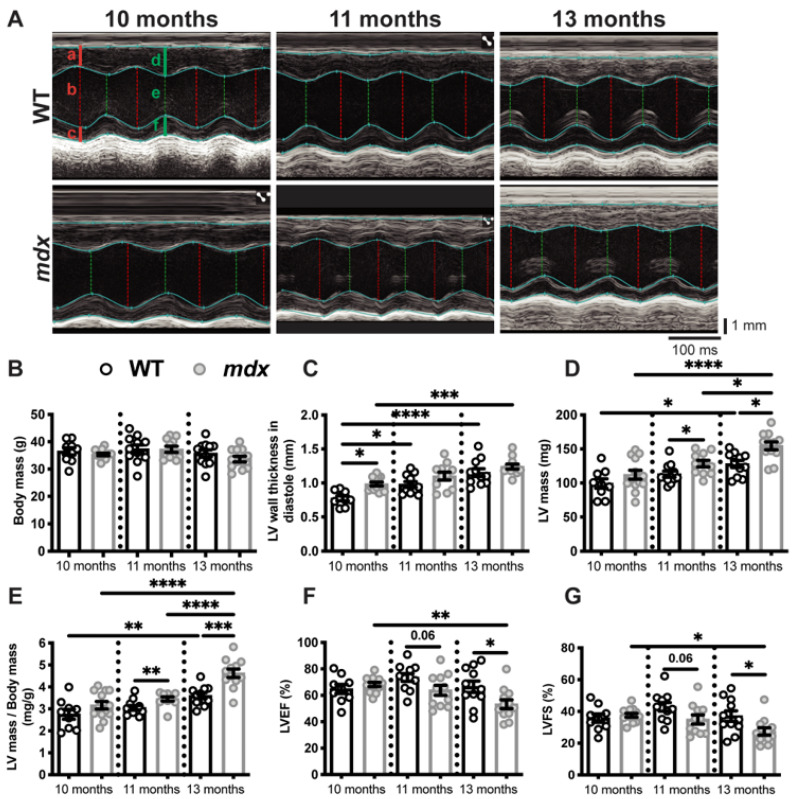
Dystrophic mice develop left ventricular hypertrophy and exhibit a decrease in cardiac function. (**A**) A Representative echocardiography of the WT and *mdx* mice at 10, 11, and 13 months of age. The vertical and horizontal scale bars represent the size in mm and the time in ms, respectively. The red lines are measured in diastole and the green lines are measured in systole. The letters a and d indicate the left ventricle (LV) anterior wall thickness, c and f show the LV posterior wall thickness, and b and e demonstrate the LV internal diameters. See Appendix A for all the data measured. The body masses (**B**), LV wall thickness in diastole (**C**), LV mass (**D**), LV mass to body mass ratio (**E**), LV ejection fraction (LVEF, (**F**)), and LV fractional shortening (LVFS, (**G**)) of the *mdx* and age-matched WT mice at 10, 11, and 13 months of age. The data are expressed as means ± SEM (* *p* < 0.05, ** *p* < 0.01, *** *p* < 0.001, and **** *p* < 0.0001). The within-age comparison used an unpaired two-tailed Student’s t-test. The across-age comparison used analysis of variance with Tukey’s post hoc test. For the WT mice, n = 9 at 10 months and n = 11 at 11 and 13 months. For the *mdx* mice, n = 13 at 10 months and n = 12 at 11 and 13 months.

**Figure 2 cells-12-01538-f002:**
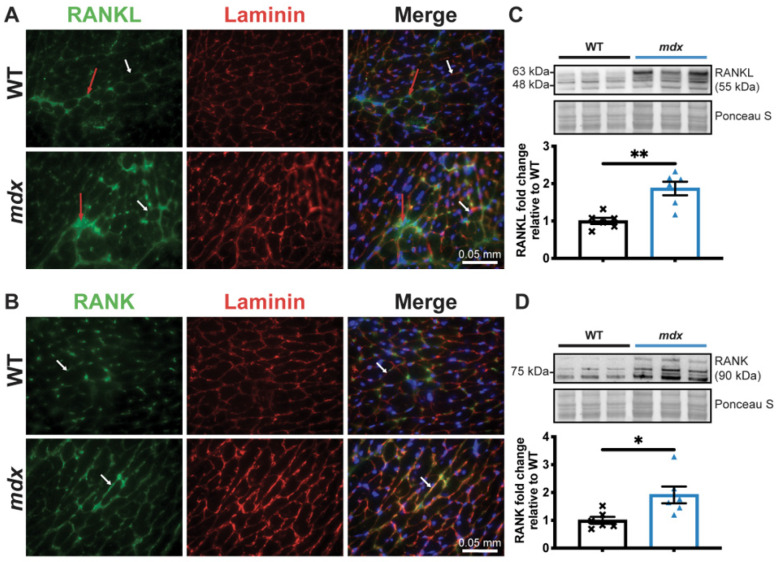
RANKL and RANK expressions are upregulated in dystrophic cardiac tissue. The heart tissues were sectioned and RANKL (**A**) and RANK (**B**) expressions were visualized using immunohistochemistry at 40× magnification. Laminin (red) and DAPI (blue) were used to label the cardiomyocyte membrane and nuclei, respectively. RANKL and RANK expression at the cell membrane and in the cardiomyocyte microenvironment are indicated by the white and red arrows, respectively (**A**,**B**). RANKL and RANK protein levels were analyzed using western blotting (**C**,**D,** respectively). The scale bar in A and B represents 0.05 mm. The results are expressed as means ± SEM (* *p* < 0.05, ** *p* < 0.01). An unpaired two-tailed Student’s *t*-test, with n = 6 for the WT and n = 6 for the *mdx*, for western blots was performed.

**Figure 3 cells-12-01538-f003:**
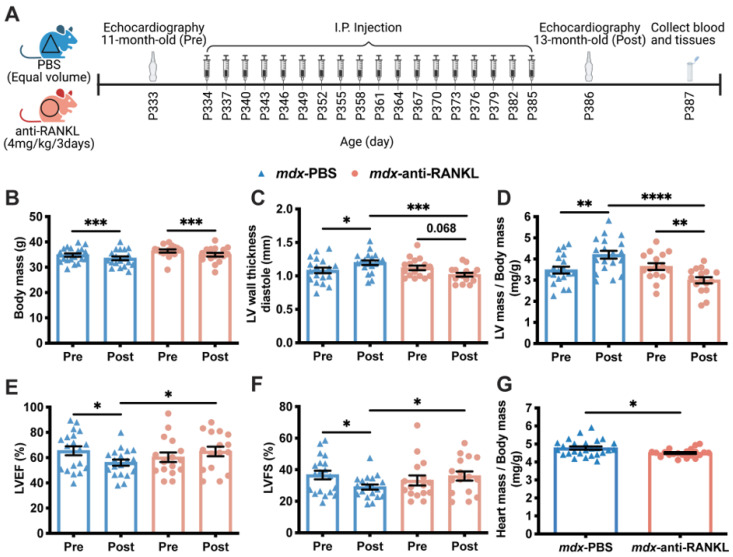
An anti-RANKL treatment improves the left ventricular morphology and maintains cardiac function in dystrophic mice. Cardiac echography was performed on the *mdx* mice at 11 months of age before the start of the treatment protocol (Pre) and at 13 months of age following the treatment period (Post, (**A**)). See Appendix A for all the data measured. The body mass (**B**), left ventricle (LV) wall thickness in diastole (**C**), LV mass to body mass ratio (**D**), left ventricular ejection fraction (LVEF, (**E**)), and fractional shortening (LVSF, (**F**)) of the *mdx* mice before and after treatment with either PBS or anti-RANKL. The post-mortem heart mass to body mass ratio of the *mdx* mice after PBS or anti-RANKL treatment (**G**). The results are expressed as means ± SEM (* *p* < 0.05, ** *p* < 0.01, *** *p* < 0.001, and **** *p* < 0.0001). The within-group comparison used a paired two-tailed Student’s *t*-test for pre- and post-treatment results. The across-group comparison used an unpaired two-tailed Student’s *t*-test for PBS and anti-RANKL treatment results. N = 19 for *mdx*-PBS and n = 16 for *mdx*-anti-RANKL.

**Figure 4 cells-12-01538-f004:**
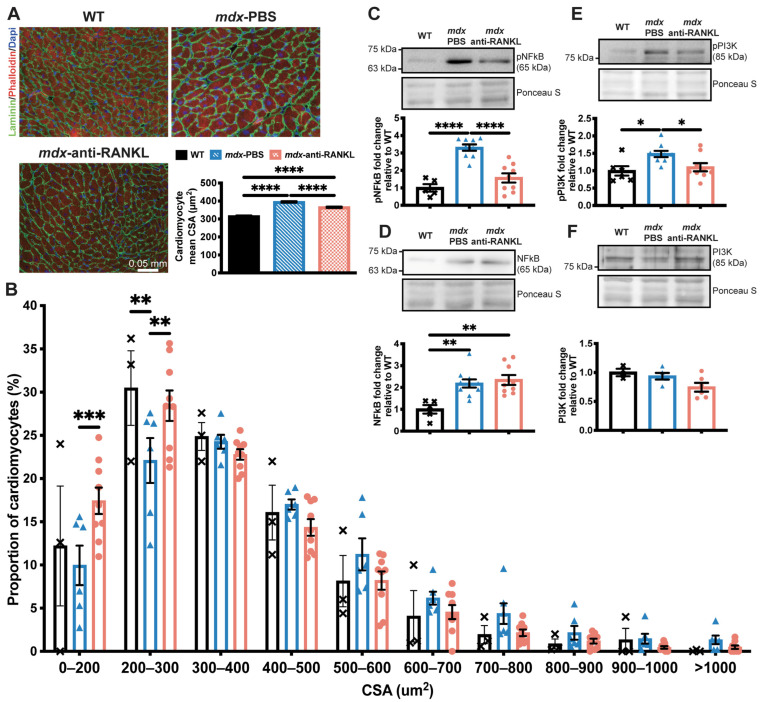
An anti-RANKL treatment reduces the cardiomyocyte surface and inhibits cardiac hypertrophy mediators in dystrophic mice. The heart tissues were sectioned and were incubated with laminin (green), rhodamine-phalloidin (red), and DAPI (blue) markers to label the cardiomyocyte membrane, F-actin filaments, and nuclei, respectively, at 20× magnification (**A**). The cardiomyocyte mean cross-sectional area (CSA) (**A**) and CSA distribution (**B**). Western blot analyses of pNFκB (**C**), NFκB (**D**), pPI3K (**E**), and PI3K (**F**) protein levels. The scale bar in A represents 0.05 mm. Results are expressed as means ± SEM (* *p* < 0.05, ** *p* < 0.01, *** *p* < 0.001, and **** *p* < 0.0001). Shown are an analysis of variance one-way ANOVA with a Tuckey correction for the CSA and western blots (**A**,**D**–**F**), and an analysis of variance two-way ANOVA with a Bonferroni correction for a distribution analysis (**B**). N = 3 for WT, n = 6 for *mdx*-PBS, and n = 9 for *mdx*-anti-RANKL for the cardiomyocyte CSA (**A**) and distribution (**B**). N = 3–6 for WT, n = 5–10 for *mdx*-PBS, and n = 5–9 for *mdx*-anti-RANKL for the western blots (**C**–**F**).

**Figure 5 cells-12-01538-f005:**
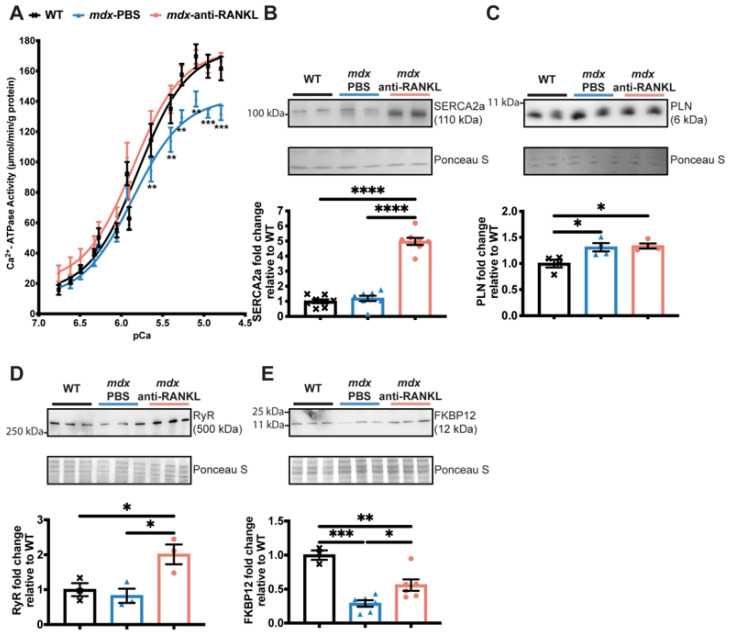
An anti-RANKL treatment increases SERCA activity and modulates intracellular calcium homeostasis regulators in the dystrophic heart. SERCA activity (**A**), western blot analyses of SERCA2a (**B**), phospholamban (PLN) (**C**), Ryanodine (RyR) (**D**), and FKBP12 (**E**) protein levels in hearts from WT, PBS-injected *mdx*, and anti-RANKL-treated *mdx* mice. Results are expressed as means ± SEM (* *p* < 0.05, ** *p* < 0.01, *** *p* < 0.001, and **** *p* < 0.0001). Shown are an analysis of variance two-way ANOVA with a Bonferroni correction (**A**) and an analysis of variance one-way ANOVA with a Tuckey correction for western blots (**B**–**E**). N = 11 for WT, *mdx*-PBS, and *mdx*-anti-RANKL for SERCA activity (**A**); n = 3–8 for WT, *mdx*-PBS, and *mdx*-anti-RANKL for Western blots (**B**–**E**).

**Figure 6 cells-12-01538-f006:**
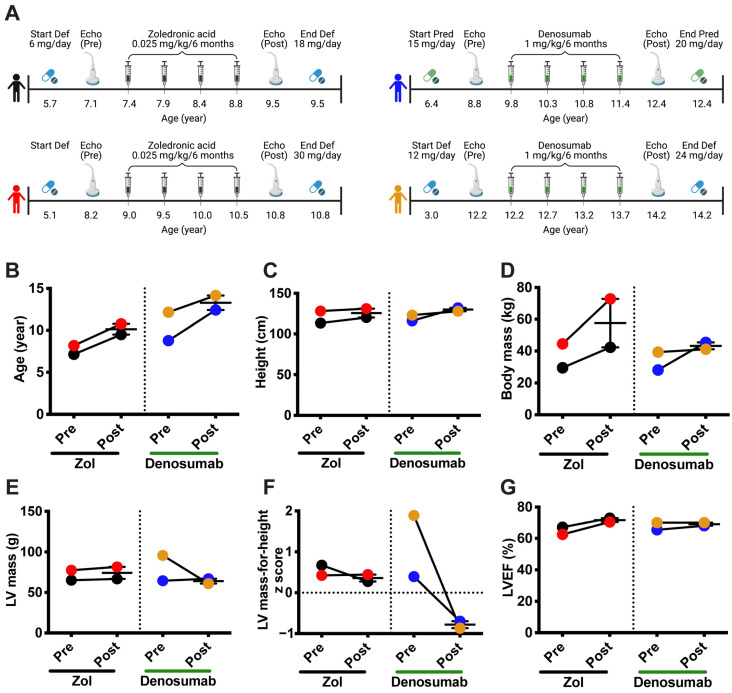
Denosumab treatment reduces left ventricular hypertrophy in patients with DMD. Four children with glucocorticoid-treated DMD, with deflazacort (Def) or prednisolone (Pred), who participated in an exploratory bone study were administered either denosumab (1 mg/kg/6 months) or a bisphosphonate (zoledronic acid: Zol, 0.025 mg/kg/6 months) for about 2 years (**A**). Cardiac morphology and function were monitored as the standard of care before (Echo Pre) and after (Echo Post) treatment (**A**). Age (**B**), height (**C**), body mass (**D**), LV mass (**E**), LV mass-for-height z score (**F**), and left ventricular ejection fraction (LVEF, (**G**)) before and after two years of treatment with Zol or Denosumab (n = 2 for the denosumab and Zol treatments).

## Data Availability

All the data generated and analyzed during the present study are included in this article.

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
