# Peer review of "RANKL Inhibition Reduces Cardiac Hypertrophy in mdx Mice and Possibly in Children with Duchenne Muscular Dystrophy"

_cells, 2023, doi:10.3390/cells12111538_

Round 1

Reviewer 1 Report

The authors of the paper "RANKL inhibition reduces cardiac hypertrophy in mdx mice and possibly in children with Duchenne Muscular dystrophy"  claims that antiRANKL treatment is able to prevent the worsening of the cardiac hypertrophy in mdx mice. The paper is well written and clear and results are explained clearly. Statistics analysis is solid even with mice N number. It also shows very promising results for clinical application. There are just a couple of observations.

1) Please specify in the material and methods which kind of injection was performed 

2) It would be nice to have also the echography B mode in the images to allow the reader to understand better the parameters

3) The immunofluorescence (Fig.2) are not really convincing. Is it possible to see a secondary only? it seems the antibody bound an aspecific. Moreover in the first images it seems background staining. On top of that, it does not seem a 40X magnification 

4) for WB analysis what did you use as a normaliser?

5) it would maybe more precise to have an intermediate time point from the beginning of the treatment and the end point.

6) have you consider testing calcium oscillation after drug exposition in cell colture to confirm your hypothesis?

7) Do you have any data on the effect of the treatment if started earlier? is there any prevention of the heart condition?

8) have you consider some more detailed analysis on isolated cardiomyocytes?

Author Response

Reviewer 1

The authors of the paper "RANKL inhibition reduces cardiac hypertrophy in mdx mice and possibly in children with Duchenne Muscular dystrophy"  claims that antiRANKL treatment is able to prevent the worsening of the cardiac hypertrophy in mdx mice. The paper is well written and clear and results are explained clearly. Statistics analysis is solid even with mice N number. It also shows very promising results for clinical application. There are just a couple of observations.

  • Please specify in the material and methods which kind of injection was performed

We have specified the kind of injection we performed in the material and methods. 

  • It would be nice to have also the echography B mode in the images to allow the reader to understand better the parameters

We have added the B mode images to the supplemental file Figure S1.

  • The immunofluorescence (Fig.2) are not really convincing. Is it possible to see a secondary only? it seems the antibody bound an aspecific. Moreover in the first images it seems background staining. On top of that, it does not seem a 40X magnification 

We have tested the secondary antibodies alone and there were no signals detected.

  • for WB analysis what did you use as a normaliser?

We used Ponceau S staining as a normalizer for each sample.

  • it would maybe more precise to have an intermediate time point from the beginning of the treatment and the end point.

We performed an echocardiography on twelve mice at 12 months of age and did not see a significant effect of the treatment. Therefore, in order to limit the stress on the mice, we decided not to do an echocardiography at an intermediate timepoint.

  • have you consider testing calcium oscillation after drug exposition in cell colture to confirm your hypothesis?

It would be interesting to study the mechanism of action of our treatment on cardiomyocyte cultures. However, the complexity of isolating and culturing dystrophic cardiomyocytes and the heterogeneity of the results which seem to significantly vary from one culture to another have made it difficult to use this approach.

  • Do you have any data on the effect of the treatment if started earlier? is there any prevention of the heart condition?

We do not have data on the effect of a treatment started well before the onset of cardiac hypertrophy. Dystrophic mice were treated right before the onset of cardiac symptoms to test whether the treatments could slow down the progression of cardiac hypertrophy. This approach also resembles to what would occur in clinic where denosumab would be given to patients with signs of cardiac hypertrophy.  

8) have you consider some more detailed analysis on isolated cardiomyocytes?

Please refer to our response indicated above.

Reviewer 2 Report

Marcadet and colleagues presented very interesting results confirming the involvement of the receptor activator of nuclear factor κB ligand in the progression of cardiomyopathy in Duchenne dystrophy. In general, the data obtained by the authors well confirm the conclusions drawn. However, I had a few design and discussion questions:

Comments:

1. In the introduction, the authors write that «A deficiency in dystrophin leads to a loss of Ca2+ homeostasis and skeletal and cardiac muscle dysfunctions».

The authors should note the generalized ion channel dysregulation in DMD, much attention is focused on calcium, but sodium and potassium ions are often not taken into account. In the latter case, this is especially important, as potassium homeostasis seems to be seriously disturbed in the heart (especially in mitochondria, which can be seen already in the early stages, when the dysregulation of calcium homestasis and, in general, the pathology of the organ is not yet expressed).

2. The authors write that «Nevertheless, cardiac dysfunctions remain poorly understood in dystrophic mdx mice, and several studies have reported conflicting results.»

For the reader's understanding, it should be noted here that this may be due to the different ages of the mice used. So, at an early stage, young mdx mice show adaptive signs, for example, improved energy metabolism (calcium uniport in mitochondria, their resistance to the MPT pore opening and improved parameters of oxidative phosphorylation), which affects the observed results. In addition, it is interesting that in female mdx mice, the pathology of the heart is more pronounced than in males, and they are often used to study the pathology of this organ.

3. For a more correct design of the experiment, the authors should evaluate the effect of therapy on healthy WT animals, this will identify possible side effects.

4. Authors need to justify the use of 4 mg/kg/3d anti-RANKL.

5. Regarding the discussion. Can the authors add something about the role of mitochondria in the observed processes? RANKL is known to be involved in the regulation of mitochondrial biogenesis and dynamics. This may be important in Duchenne dystrophy, which shows severe defects in the structure and function of these organelles.

Author Response

Reviewer 2

Marcadet and colleagues presented very interesting results confirming the involvement of the receptor activator of nuclear factor κB ligand in the progression of cardiomyopathy in Duchenne dystrophy. In general, the data obtained by the authors well confirm the conclusions drawn. However, I had a few design and discussion questions:

Comments:

  1. In the introduction, the authors write that «A deficiency in dystrophin leads to a loss of Ca2+ homeostasis and skeletal and cardiac muscle dysfunctions». 

The authors should note the generalized ion channel dysregulation in DMD, much attention is focused on calcium, but sodium and potassium ions are often not taken into account. In the latter case, this is especially important, as potassium homeostasis seems to be seriously disturbed in the heart (especially in mitochondria, which can be seen already in the early stages, when the dysregulation of calcium homestasis and, in general, the pathology of the organ is not yet expressed).

We have added few sentences in the introduction on ion channels dysregulated in DMD.

  1. The authors write that «Nevertheless, cardiac dysfunctions remain poorly understood in dystrophic mdx mice, and several studies have reported conflicting results.»

For the reader's understanding, it should be noted here that this may be due to the different ages of the mice used. So, at an early stage, young mdx mice show adaptive signs, for example, improved energy metabolism (calcium uniport in mitochondria, their resistance to the MPT pore opening and improved parameters of oxidative phosphorylation), which affects the observed results. In addition, it is interesting that in female mdx mice, the pathology of the heart is more pronounced than in males, and they are often used to study the pathology of this organ.

So far, we have found only one study that compares cardiac function in male and female mdx mice (Bostick, Yue, and Duan 2010). The study reported that only the ejection fraction, systolic and diastolic volume were worse in female dystrophic mice when compared to their male counterparts.  The other parameters such as stroke volume, cardiac output, and ECG parameters were either similar between both sexes or worse in male mice. According to the literature, female mice are less affected than males, especially since estrogen has a protective role on the cardiovascular function (Luo and Kim 2016; Cavasin et al. 2004). Most of the published studies investigating dystrophic heart function used male mdx mice.

  1. For a more correct design of the experiment, the authors should evaluate the effect of therapy on healthy WT animals, this will identify possible side effects. 

Anti-RANKL may potentially increase bone mass in WT mice. These mice may have too much bone which is a disease called osteopetrosis. Therefore, we do not treat WT type mice with anti-RANKL. Denosumab, a human anti-RANKL, had already been tested on human population and no significant adverse cardiac effect has been reported.

  1. Authors need to justify the use of 4 mg/kg/3d anti-RANKL.

We have justified the use of 4mg/kg/3d in the Material and Methods section.

  1. Regarding the discussion. Can the authors add something about the role of mitochondria in the observed processes? RANKL is known to be involved in the regulation of mitochondrial biogenesis and dynamics. This may be important in Duchenne dystrophy, which shows severe defects in the structure and function of these organelles.

It is an interesting suggestion. However, since we do not have data on the effects of anti-RANKL on mitochondria function, we believe that it is a bit hasty and speculative to address the role of mitochondria in the present study.

References

Bostick, Brian, Yongping Yue, and Dongsheng Duan. 2010. “Gender Influences Cardiac Function in the Mdx Model of Duchenne Cardiomyopathy.” Muscle & Nerve 42 (4): 600–603. https://doi.org/10.1002/mus.21763.

Cavasin, Maria A., Zhenyin Tao, Shreevidya Menon, and Xiao-Ping Yang. 2004. “Gender Differences in Cardiac Function during Early Remodeling after Acute Myocardial Infarction in Mice.” Life Sciences 75 (18): 2181–92. https://doi.org/10.1016/j.lfs.2004.04.024.

Luo, Tao, and Jin Kyung Kim. 2016. “The Role of Estrogen and Estrogen Receptors on Cardiomyocytes: An Overview.” Canadian Journal of Cardiology 32 (8): 1017–25. https://doi.org/10.1016/j.cjca.2015.10.021.

Round 2

Reviewer 2 Report

The authors answered my questions and comments. The work may be accepted.